

# Determining novel candidate anti-hepatocellular carcinoma drugs using interaction networks and molecular docking between drug targets and natural compounds of SiNiSan

Qin Zhang[1,*], Zhangying Feng[2,*], Mengxi Gao[1] and Liru Guo[1]

[1] The Fourth Hospital of Hebei Medical University, Department of General Medicine, Shijiazhuang, Hebei, China

[2] The Fourth Hospital of Hebei Medical University, Department of Clinical Pharmacology, Shijiazhuang, Hebei, China

[*] These authors contributed equally to this work.

Corresponding author
Liru Guo, qkglrdoctor@126.com

## ABSTRACT

**Background**. SiNiSan (SNS) is an ancient traditional Chinese medicine (TCM) used to treat liver and spleen deficiencies. We studied the unique advantages of using SNS to treat hepatocellular carcinoma (HCC) with multiple components and targets to determine its potential mechanism of action.

**Methods**. The active compounds from the individual herbs in the SNS formula and their targets were mined from Traditional Chinese Medicine Systems Pharmacology Database (TCMSP). HCC-associated targets were collected from the TCGA and GEO databases and samples were collected from patients with stage III hepatocellular carcinoma. A compound-disease target network was constructed, visualized, and analyzed using Cytoscape software. We built a protein-protein interaction (PPI) network using the String database. We enriched and analyzed key targets using GSEA, GO, and KEGG in order to explore their functions. Autodock software was used to simulate the process of SNS molecules acting on HCC targets.

**Results**. A total of 113 candidate compounds were taken from SNS, and 64 of the same targets were chosen from HCC and SNS. The predominant targets genes were PTGS2, ESR1, CHEK1, CCNA2, NOS2 and AR; kaempferol and quercetin from SNS were the principal ingredients in HCC treatment. The compounds may work against HCC due to a cellular response to steroid hormones and histone phosphorylation. The P53 signaling pathway was significantly enriched in the gene set GSEA enrichment analysis and differential gene KEGG enrichment analysis.

**Conclusions**. Our results showed that the SNS component has a large number of stage III HCC targets. Among the targets, the sex hormone receptors, the AR and ESR1 genes, are the core targets of SNS component and the most active proteins in the PPI network. In addition, quercetin, which has the most targets, can act on the main targets (BAX, CDK1, CCNB1, SERPINE1, CHEK2, and IGFBP3) of the P53 pathway to treat HCC.

## INTRODUCTION

Hepatocellular carcinoma (HCC) is an extremely malignant type of tumor that accounts for the majority of primary liver cancer cases worldwide. The increased incidence rate of HCC makes it the leading cause of death among patients with cirrhosis (*Forner, Reig & Bruix, 2018*). Liver cancer is the sixth most common neoplasm and the fourth leading cause of cancer death according to the GLOBOCAN 2018 report issued by the International Agency for Research (IARC) (*Ferlay et al., 2019*).

HCC develops in stages and is caused by a combination of factors. Determining the molecular mechanisms of HCC is complicated due to the physiological functions of the liver (*Pan, Fu & Huang, 2011*). The following mechanisms have been confirmed: activation of proto-oncogenes, such as N-ras and HBVx (*Cheng et al., 2014*; *Kudo, 2011*); inactivation of the tumor suppressor genes, such as p53, Rb, p21 and PTEN (*Sandra & Jean-Charles, 2020*; *Maheshkumar et al., 2019*); abnormal activation of multiple molecular signaling pathways; and the expression of HCC-associated proteins. HCC treatments focus on increasing survival while maintaining a high quality of life. The treatments include surgery, interventional therapy, chemotherapy, and combination therapy. Surgical treatments include hepatectomy, liver transplantation, and local ablative therapies. However, these are only used for early-stage liver cancer and tumors that are localized and not metastasized (*Villanueva, 2019*). Vascular interventional therapy mainly consists of transarterial chemoembolization (TACE) and transcatheter arterial infusion (TAI), which can effectively control the growth of HCC cells. However, there is no standardized staging or guidelines for interventional therapy ischemia, and hypoxia in tumor tissues after TACE therapy may lead to the increase of hypoxia-inducible factors, resulting in the high expression of VEGF in residual tumors, and eventually causing cancer recurrence and metastasis. Chemotherapy is an important option for HCC treatment but is limited by its toxic side effects. An effective systematic treatment is required for patients with terminal-stage liver cancer or patients with severe underlying disease to act on multiple mechanisms and control tumor progression (*Grandhi et al., 2016*). Tumor staging is a crucial step in the treatment of HCC. TNM (Tumour, Node, Metastasis) and BCLC (The Barcelona Clinic Liver Cancer) are currently the most widely used evaluation systems, combined with the HCC study published in the Lancet in 2018 and the 2016 version of the AASLD (American Association for the Study of Liver Diseases) guideline (*Forner, Reig & Bruix, 2018*; *Heimbach et al., 2018*). Patients with early stage HCC that tumor diameters ≤5 cm (T1, T2 or BCLC0, BCLCA) can benefit from resection, transplantation, and ablation treatments, for advanced HCC, the selection of treatment type depends on the extent of invasion of macrovascular, metastasis, and the physical condition of the patient, systemic treatment and the best supportive care may be the best choice, and this is exactly what we hope TCM achieves. Therefore, this study selected stage III hepatocellular carcinoma as samples for follow-up research.

Traditional Chinese medicine (TCM) refers to medicines used in China for thousands of years; they are typically derived from different natural medicines and herbal products and have made significant contributions to human health (*Tao et al., 2015*; *Hao et al., 2017*;

*Cheng, Qi & Wang, 2019*; *Xiao-Cong et al., 2018*). Advances in analytical technologies and methodologies have accelerated the research of TCM, and the role of TCM in anti-tumor alternative therapies has received increasing attention (*Lee et al., 2019*). Studies have confirmed that TCM is effective in improving the survival rate of patients with breast and lung cancers (*Lee et al., 2014*; *Hsiao & Liu, 2010*; *Liao et al., 2017*). SiNiSan (SNS) is a basic TCM formulation and was first introduced in the Treatise on Febrile Disease by Zhang Zhongjing, a famous physician in the late eastern Han Dynasty. SNS consists of four herbs: Gancao (Radix Glycyrrhizae), Chaihu (Radix Bupleuri), Zhishi (Fructus Aurantii Immaturus), and Baishao (Radix Paeoniae Alba). It is used for treating liver stagnation and spleen deficiency, improving disorders of the digestive system, and alleviating depression (*Jiang et al., 2003*). Studies have confirmed that SNS can affect the invasiveness and metastatic potential of hepatocellular carcinoma cells by inhibiting the phosphorylation of extracellular signal-related kinases and c-Jun N-terminal kinase signaling pathway (*Hung et al., 2015*). The mechanism is unclear, which has limited its clinical applications, thus, the therapeutic effects of SNS on hepatocellular carcinoma needs to be determined. Network pharmacology (NP) is an emerging drug research strategy that places the effects of drugs on diseases in a complex biological network and uses related gene databases to determine the mechanism of action of the drug compounds (*Zhang et al., 2019*). NP has accelerated pharmacological development and improved our understanding of mechanisms of drug action by establishing a multi-layer network of disease-phenotype-gene-drug (*Xiao-Ming & Chun-Fu, 2015*) and is an important application for TCM research.

We downloaded the active pharmaceutical ingredients and target genes of SNS from the TCMSP database and obtained the disease genes of stage III hepatocellular carcinoma (HCC) from the TCGA and GEO databases. Gene Set Enrichment Analysis (GSEA) enrichment analysis was conducted on the gene sets of the two databases, and differential analysis was conducted to screen out the HCC genes with significant differences. The compound-disease target network was obtained from the intersection of the SNS targets and the HCC differential genes. The possible mechanisms and pathways of SNS were studied by analyzing the active compounds of SNS in the network, performing GO and KEGG enrichment analysis on the intersected genes, and constructing a protein interaction network. Finally, we used molecular docking to simulate the process of SNS active molecules acting on target HCC genes. We investigated the bioactive compounds, crossing targets, and possible mechanisms of SNS on hepatocellular carcinoma using a network pharmacology strategy to provide new theoretical treatments.

## MATERIALS & METHODS

### Screening of bioactive compounds and targets in SiNiSan

The individual herbs and active components (Radix Glycyrrhizae, Radix Bupleuri, Fructus Aurantii Immaturus and Radix Paeoniae Alba) in SNS were mined from the Traditional Chinese Medicine Systems Pharmacology Database (TCMSP: http://tcmspw.com/tcmsp.php/). This database contains the active ingredients of Chinese medicine and their corresponding targets (*Yang et al., 2019*). Herbal medicines are screened

by TCMSP based on absorption, distribution, metabolism, and excretion (ADME) and includes factors like oral bioavailability (OB), drug-likeness (DL), and P450 (*Lee et al., 2019*). OB represents the speed and degree of drug absorption in the circulation and DL represents the similarity between the herbal ingredients and specific medicines, suggesting that herbs may be used as therapeutic agents. Refer to TCMSP's platform research and related traditional Chinese medicine research in the past (*Jinlong et al., 2014*; *Xue et al., 2012*), we selected OB and DL as conditions and set the filter criteria to OB ≥30% and DL ≥0.18 for potential bioactive compounds by Strawberry-perl software (https://www.perl.org, ver.5.30.1.1).

## Prediction and Gene Set Enrichment Analysis (GSEA) enrichment analysis of stage III hepatocellular carcinoma targets

HCC-associated targets were acquired from The Cancer Genome Atlas (TCGA) and Gene Expression Omnibus (GEO). TCGA is a collaborative database from the National Cancer Institute (NCI) and the National Human Genome Institute (NHGRI), which is responsible for a large amount of clinical and genomic data for various cancer types. The GEO database is an international public gene expression database affiliated with NCBI that covers sufficient disease-differential genes profiling by array (*Barrett, Wilhite & Ledoux, 2013*). TCGA data were obtained from the GDC Data Portal (https://portal.gdc.cancer.gov/), select the TCGA-LIHC (Liver Hepatocellular Carcinoma) project in the liver cancer classification, pick and summarize the stage III hepatocellular carcinoma cases as the disease Group, and the normal adjacent tissues in HCC cases as the normal group. Similarly, search for HCC data in the GEO database (https://www.ncbi.nlm.nih.gov/) with "hepatocellular carcinoma" and "normal" as keywords, and set the conditions as Homo sapiens to obtain a set of HCC gene expression profiles (GSE101685) and its RNA chip annotation files (GPL570), then based on the annotation file, use Strawberry-perl-5.30.1.1 software to convert the chip probe ID into gene symbol, and selected the HCC samples of stage III as the disease group, and the normal tissue samples as the normal group. GSEA calculated the results of the overall pathway or function enrichment through enrichment analysis of the gene sets that did not undergo differential analysis nor did it interfere with artificially set thresholds, which is to determine whether the predefined gene set can show a consistent enrichment trend in overall genes include the disease group and the normal group. In order to perform GSEA analysis, the above-mentioned GEO and TCGA gene data were sorted into a matrix file which includes the disease group and the normal group. We selected the Canonical pathways of the C2 curated gene sets (c2.cp.pid.v7.1) in the Molecular Signatures Database (MSigDB) and analyzed them using GSEA software (ver 4.0.3).

## Differential analysis of stage III hepatocellular carcinoma targets and the building of a compound-disease target network

Analyzing the HCC sample genes and normal sample genes from TCGA and GEO, and get the differentially expressed genes (DEGs) that are significantly different from the normal group in the disease group. DEGs for stage III HCC were obtained by limma package in R software (https://www.r-project.org, version 3.6.2) based on two criteria ($|logFC|>1$ and adj.P Value<0.05). The duplicates were removed after obtaining the differential genes from

the TCGA and GEO databases. We determined the intersection of the SNS bioactive drug targets and DEGs of HCC using Strawberry-perl-5.30.1.1 software to obtain the targets for compound-disease network. The compound-disease target network was built, visualized, and analyzed using Cytoscape software (https://cytoscape.org, version 3.7.2). The nodes in the network indicated the HCC target genes and the SNS bioactive compounds. The edges indicated their interactions.

### Protein–protein interaction (PPI) network

The protein–protein interaction (PPI) network was constructed using the string database (https://string-db.org/) based on crossing targets that determined the interactions between gene regulatory proteins. The string database contains a large amount of protein interaction information from experiments or other databases and can predict the mode of action between HCC-related proteins. We entered 64 intersection HCC genes, selected the species as Homo sapiens, and set the minimum required interaction score to 0.4 (default) to set up a PPI network. The active protein interaction sources included text mining, experiments, databases, co-expressions, neighborhoods, gene fusions, and co-occurrence. We imported the network to Cytoscape for node and connection analyses.

### GO and KEGG pathway enrichment analysis

We conducted Gene Ontology (GO) and Kyoto Encyclopedia of Genes and Genomes (KEGG) enrichment analysis of 64 HCC target genes to clarify the biological functions and pathways of SNS compounds for treating HCC. GO enrichment analysis included the biological process (BP) and cell compounds (CC). GO and KEGG enrichment analysis ($P \leq 0.05$, $Q \leq 0.05$) were conducted using the bioconductor package in R software (ver3.6.2).

### Molecular docking to simulate the core target-compound binding

We downloaded the PDB file of targets from the RCSB PDB website (https://www.rcsb.org/#Category-search) and the MOL2 file of the compound ligand from the ZINC database (http://zinc.docking.org/) and selected a target related to HCC and performed molecular docking with the corresponding compound. We used the AutoDock software (ver4.2.6) to dehydrate and hydrogenate the macromolecules. We calculated the charge of the target macromolecular protein, determined the rotation bond of the ligand, selected the macromolecule, set the box, ran Autogrid to make the macromolecular crystal structure, and ran Autodock to create a rigid docking of the macromolecules and ligands. The docking algorithm was set as the local search parameter using Python 2.5 and we used Pymol software (ver2.3.4) to perform the visualization of the docking result.

## RESULTS

### Bioactive compounds and targets in SiNiSan

A total of 779 compounds and 6,984 targets in SNS were collected from TCMSP. We obtained 92 bioactive compounds and 1,769 targets in Radix Glycyrrhizae, 17 bioactive compounds and 123 targets in Radix Bupleur, 22 bioactive compounds and 348 targets in

Fructus Aurantii Immaturus, 13 bioactive compounds and 306 targets in Radix Paeoniae Alba with 21 repeat compounds after screening by OB ≥30% and DL ≥ 0.18. 113 compounds were identified as potential bioactive molecules for further study.

## Hepatocellular carcinoma-associated targets and GSEA enrichment analysis results

The genetic data from the TCGA database included 47 stage III hepatocellular tumor samples and seven normal adjacent tissue samples. The dataset (GSE101685) selected from the GEO database had 24 samples including eight control (normal) samples and 16 test (stage III HCC) samples. The results of GSEA enrichment analysis of the HCC gene sets in the two databases showed that the stage III hepatocellular carcinoma group from TCGA was significantly enriched in 82 pathways (nominal $p$ value <0.05 and FDR <25%), sorted by their normalized enrichment score (NES). The pathways with the highest enrichment levels are shown in Fig. 1. The stage III hepatocellular carcinoma group from GEO was significantly enriched in nine pathways (nominal $p$ value <0.05 and FDR <25%), as shown in Fig. 2. The enrichment bar plot of the genes in the two databases are shown in Fig. 3. Based on the above results, all nine pathways enriched in the HCC gene set from GEO are also enriched in the TCGA data. HCC genes were significantly enriched in the ATR, FANCONI, AURORA_B, PLK1, ARF3, MTOR4, P53_REGULATION, LKB1, ATM, and E2F pathways. All of these pathways are upregulated in HCC.

A total of 3,569 differential genes were obtained after differential analysis including 1,614 up-regulated genes and 955 down-regulated genes from TCGA. We obtained 1,078 differential genes, including 370 up-regulated genes and 708 down-regulated genes from GEO. Differential expression genes of TCGA and GEO are shown in a volcano map (Fig. 4).

## Analysis of compound-disease target network

A total of 3,101 HCC differential genes were obtained after synthesizing the differential genes of the two databases and removing the duplicates. Sixty-four HCC target genes were obtained through the intersection of SNS targets and HCC differential genes. We built a visualized compound-disease target network by combining these with 113 bioactive compounds that had been previously obtained (Fig. 5). A total of 177 nodes and 610 edges are shown in the network. Most SNS molecules can act on multiple HCC targets, the most predominant of which is quercetin (MOL000098) with 40 edges, followed by kaempferol (MOL000422) with 17 edges. Table 1 shows the 30 most prevalent ingredients in SNS prescriptions that act on the largest number of HCC targets. PTGS2 (106 edges), ESR1 (84 edges), AR (74 edges), NOS2 (74 edges), CCNA2 (53 edges) and CHEK1 (50 edges) are the genes regulated by at least 1 compound. The target gene linkage is >50 degrees, indicating that these targets might be a key target for SNS treatment of HCC.

## Analysis of protein-protein interactions (PPI) network

The protein interactions of HCC differential genes extracted from the human genome were shown in the PPI network (Fig. 6). Each node in the network diagram represents a target gene, the connection between the nodes indicates that there is an interaction relationship, and the color of the nodes changing from blue to red represents the degree of the nodes

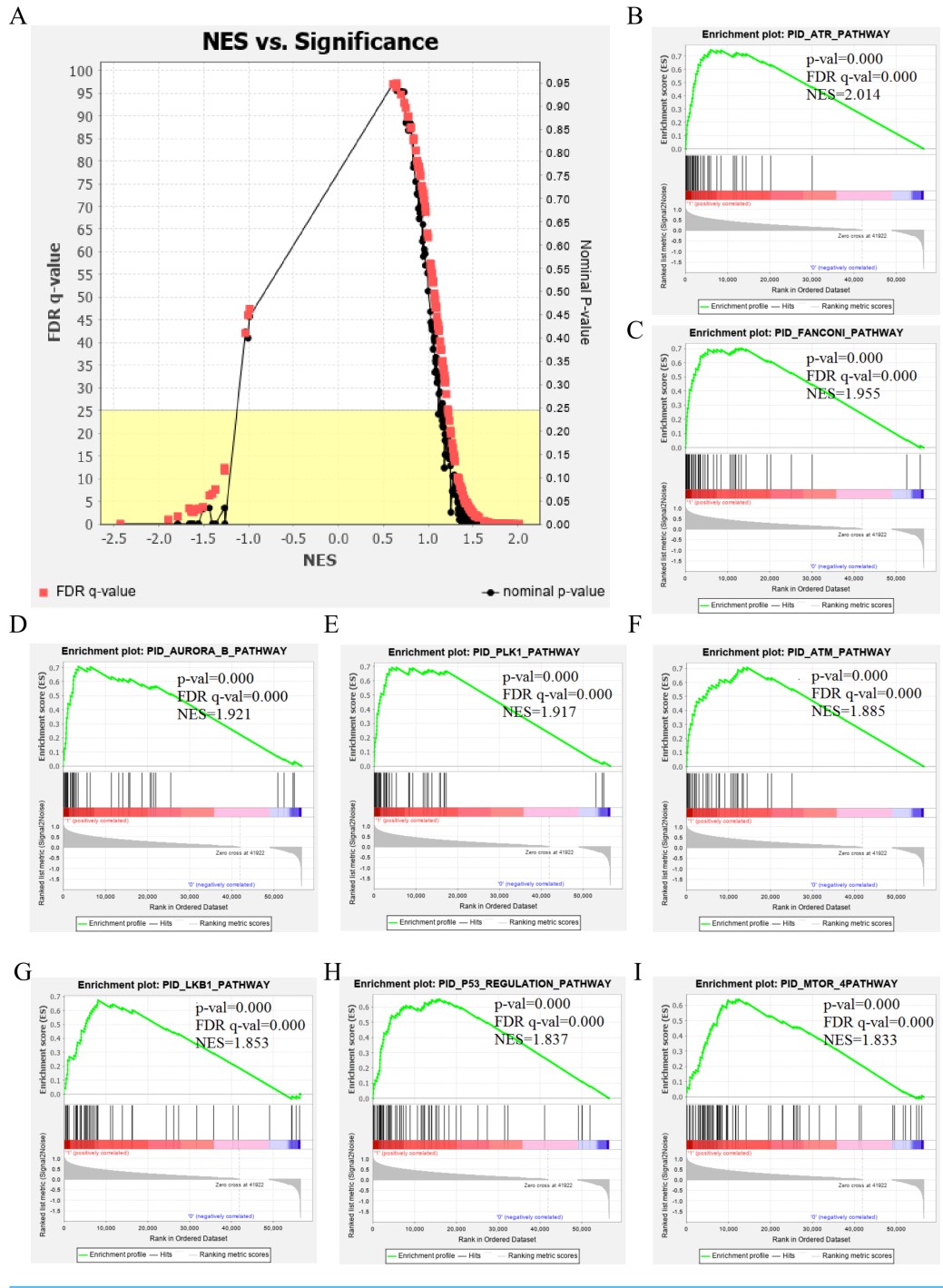

**Figure 1** **The results of GSEA enrichment analysis of stage III HCC genes from TCGA database.** (A) The normalized enrichment score (NES) and significance analysis of C2 curated gene sets (c2.cp.pid.v7.1) of GSEA; (B–I) The top eight significantly enriched pathways (*p* value < 0.05 and FDR < 25%), namely: ARF pathway, ATM pathway, P53 REGULATION pathway, ATR pathway, AURORA A pathway, E2F pathway, FANCONI pathway and FOXM1 pathway.

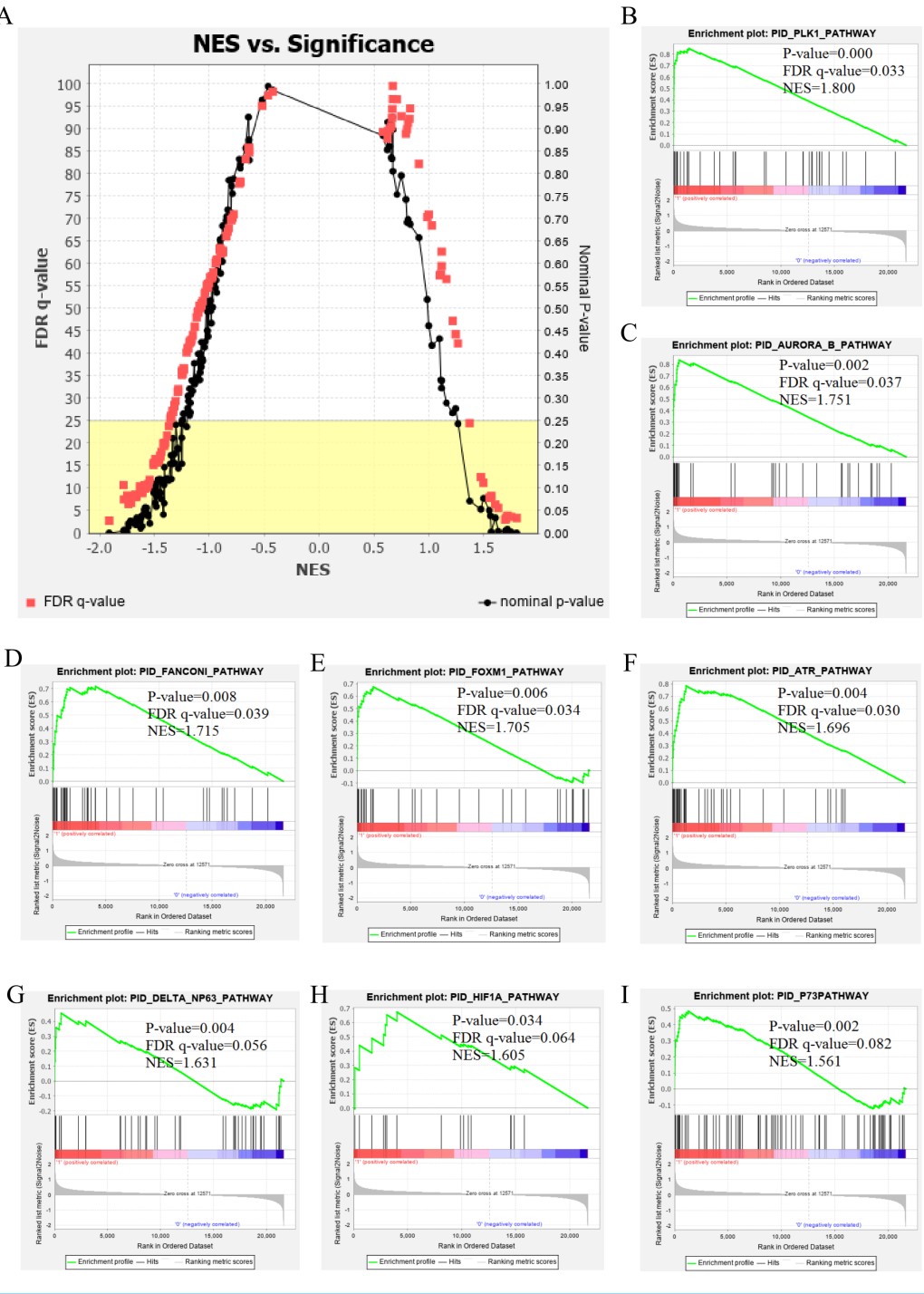

**Figure 2** **The results of GSEA enrichment analysis of stage III HCC genes from GEO database.** (A) The normalized enrichment score (NES) and significance analysis of C2 curated gene sets (c2.cp.pid.v7.1) of GSEA; (B–I) The top 8 significantly enriched pathways (*p* value < 0.05 and FDR < 25%), namely: ATR pathway, AURORA B pathway, DELTA NP63 pathway, FANCONI pathway, FOXM1 pathway, HIF1A pathway, P73 pathway and PLK1 pathway.

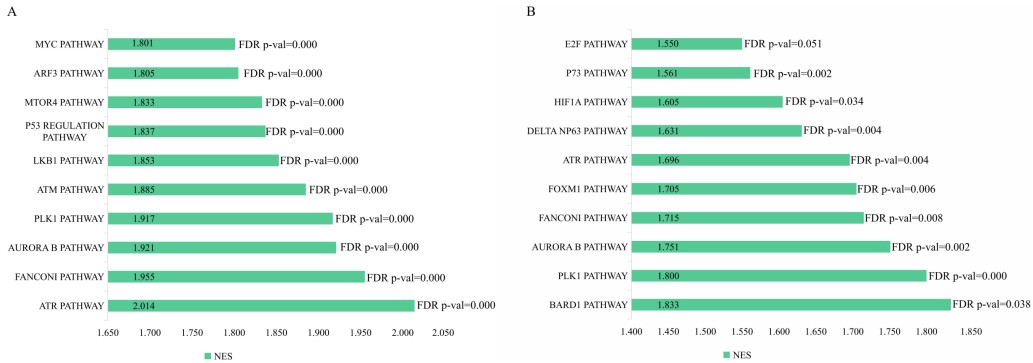

**Figure 3   The enriched pathway of GSEA analysis based on HCC targets from TCGA and GEO database.** (A) GSEA enrichment analysis of stage III HCC genes from TCGA database. (B) GSEA enrichment analysis of stage III HCC genes from GEO database.

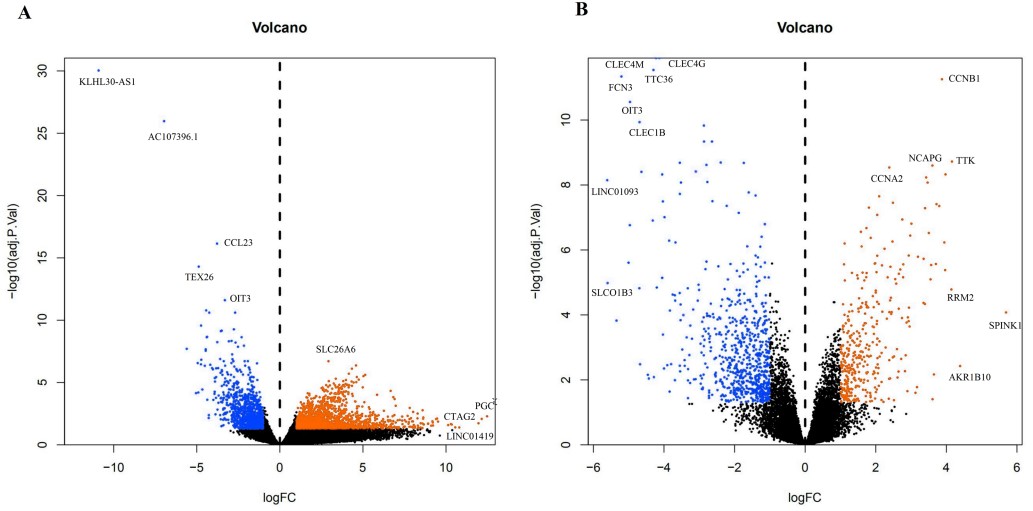

**Figure 4   The diffferentially expressed genes(DEGs) between normal and stage III HCC tissues.** The top genes were marked (A) DEGs from TCGA database; (B) DEGs from GEO database.

changing from low to high. The overall network included 64 nodes and 343 edges (PPI enrichment $p$-value: $<1.0e-16$), the average node degree was 10.7 and the genes with the highest node degrees are: ESR1 (31 edges), MYC (30 edges), JUN (30 edges), FOS (25 edges), CAT (24 edges), and AR (22 edges).

## GO and KEGG pathway enrichment analysis

The results of GO enrichment analysis of the 64 HCC target genes showed that the effect of SNS on HCC may be mainly associated with the response to factors including metalion, cadmiumion, oxidative stress, steroid hormone, histone phosphorylation, cyclin-dependent protein kinase holoenzyme complex, RNA polymerase II transcription factor complex, and the chromosomal region. The distribution of enrichment results is shown in Fig. 7.

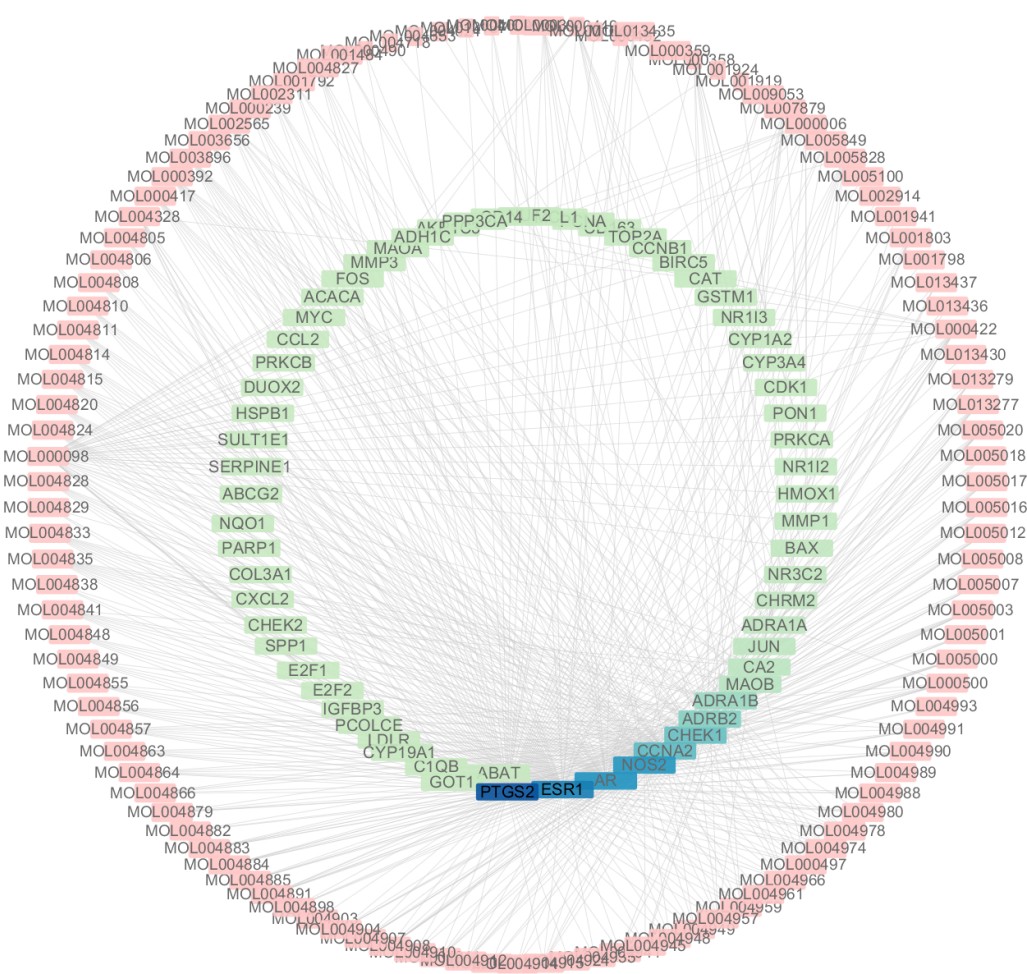

**Figure 5  The targets of each candidate compound in SiNiSan on stage III HCC.** The outer circle represents the SNS component and the inner circle represents the HCC target, darkened nodes indicate key genes for drug targeting.

According to KEGG enrichment analysis of HCC target genes, the major pathways for herbal interventions in HCC are the p53 signaling pathway (hsa04115), calcium signaling pathway (hsa04020), NF-kappa B signaling pathway (hsa04064), and the HIF-1 signaling pathway (hsa04066). The enrichment results of KEGG pathways are shown in Table 2.

## Docking results of compound and core target

Our results showed that quercetin can act on the greatest number of HCC targets. Therefore, we docked the core targets involved in the component disease network (AR, BAX, CHEK2, ESR1, NOS2, PTGS2) with quercetin (Fig. 5). We downloaded the PDB files of the HCC targets and the Mol2 files of the ligand quercetin (ZINC6484693) for docking. The quercetin structure has 6 rotatable bonds and the hydrogen bond binding energy, total internal energy, and electrostatic energy are distinct due to its position and the different positions of the amino acid fragments of the target protein. The docking results of the
**Table 1  Active ingredients and parameters of SiNiSan.**

| ID | Herb | Compound | OB(%) | DL |
|---|---|---|---|---|
| MOL000358 | Baishao | beta-sitosterol | 36.91 | 0.75 |
| MOL000492 | Baishao | (+)-catechin | 54.83 | 0.24 |
| MOL001919 | Baishao | (3S,5R,8R,9R,10S,14S)-3,17-dihydroxy-4,4,8,10,14-pentamethyl-2,3,5,6,7,9-hexahydro-1H-cyclopenta[a]phenanthrene-15,16-dione | 43.56 | 0.53 |
| MOL001924 | Baishao | paeoniflorin | 53.87 | 0.79 |
| MOL000449 | Chaihu | Stigmasterol | 43.83 | 0.76 |
| MOL000490 | Chaihu | petunidin | 30.05 | 0.31 |
| MOL001645 | Chaihu | Linoleyl acetate | 42.1 | 0.2 |
| MOL004598 | Chaihu | 3,5,6,7-tetramethoxy-2-(3,4,5-trimethoxyphenyl)chromone | 31.97 | 0.59 |
| MOL004609 | Chaihu | Areapillin | 48.96 | 0.41 |
| MOL004653 | Chaihu | (+)-Anomalin | 46.06 | 0.66 |
| MOL004718 | Chaihu | $\alpha$-spinasterol | 42.98 | 0.76 |
| MOL013187 | Chaihu | Cubebin | 57.13 | 0.64 |
| MOL000239 | Gancao | Jaranol | 50.83 | 0.29 |
| MOL000392 | Gancao | formononetin | 69.67 | 0.21 |
| MOL000417 | Gancao | Calycosin | 47.75 | 0.24 |
| MOL000497 | Gancao | licochalconea | 40.79 | 0.29 |
| MOL000500 | Gancao | Vestitol | 74.66 | 0.21 |
| MOL001484 | Gancao | Inermine | 75.18 | 0.54 |
| MOL001792 | Gancao | DFV | 32.76 | 0.18 |
| MOL002311 | Gancao | Glycyrol | 90.78 | 0.67 |
| MOL002565 | Gancao | Medicarpin | 49.22 | 0.34 |
| MOL003656 | Gancao | Lupiwighteone | 51.64 | 0.37 |
| MOL003896 | Gancao | 7-Methoxy-2-methyl isoflavone | 42.56 | 0.2 |
| MOL004805 | Gancao | (2S)-2-[4-hydroxy-3-(3-methylbut-2-enyl)phenyl]-8,8-dimethyl-2,3-dihydropyrano[2,3-f]chromen-4-one | 31.79 | 0.72 |
| MOL004806 | Gancao | euchrenone | 30.29 | 0.57 |
| MOL004808 | Gancao | glyasperin B | 65.22 | 0.44 |
| MOL004810 | Gancao | glyasperin F | 75.84 | 0.54 |
| MOL004811 | Gancao | Glyasperin C | 45.56 | 0.4 |
| MOL004814 | Gancao | Isotrifoliol | 31.94 | 0.42 |
| MOL004815 | Gancao | (E)-1-(2,4-dihydroxyphenyl)-3-(2,2-dimethylchromen-6-yl)prop-2-en-1-one | 39.62 | 0.35 |
| MOL004820 | Gancao | kanzonols W | 50.48 | 0.52 |
| MOL004824 | Gancao | (2S)-6-(2,4-dihydroxyphenyl)-2-(2-hydroxypropan-2-yl)-4-methoxy-2,3-dihydrofuro[3,2-g]chromen-7-one | 60.25 | 0.63 |
| MOL004827 | Gancao | Semilicoisoflavone B | 48.78 | 0.55 |
| MOL004828 | Gancao | Glepidotin A | 44.72 | 0.35 |

**Table 1** (*continued*)

| ID | Herb | Compound | OB(%) | DL |
|---|---|---|---|---|
| MOL004829 | Gancao | Glepidotin B | 64.46 | 0.34 |
| MOL004833 | Gancao | Phaseolinisoflavan | 32.01 | 0.45 |
| MOL004835 | Gancao | Glypallichalcone | 61.6 | 0.19 |
| MOL004838 | Gancao | 8-(6-hydroxy-2-benzofuranyl)-2,2-dimethyl-5-chromenol | 58.44 | 0.38 |
| MOL004841 | Gancao | Licochalcone B | 76.76 | 0.19 |
| MOL004848 | Gancao | licochalcone G | 49.25 | 0.32 |
| MOL004849 | Gancao | 3-(2,4-dihydroxyphenyl)-8-(1,1-dimethylprop-2-enyl)-7-hydroxy-5-methoxy-coumarin | 59.62 | 0.43 |
| MOL004855 | Gancao | Licoricone | 63.58 | 0.47 |
| MOL004856 | Gancao | Gancaonin A | 51.08 | 0.4 |
| MOL004857 | Gancao | Gancaonin B | 48.79 | 0.45 |
| MOL004863 | Gancao | 3-(3,4-dihydroxyphenyl)-5,7-dihydroxy-8-(3-methylbut-2-enyl)chromone | 66.37 | 0.41 |
| MOL004864 | Gancao | 5,7-dihydroxy-3-(4-methoxyphenyl)-8-(3-methylbut-2-enyl)chromone | 30.49 | 0.41 |
| MOL004866 | Gancao | 2-(3,4-dihydroxyphenyl)-5,7-dihydroxy-6-(3-methylbut-2-enyl)chromone | 44.15 | 0.41 |
| MOL004879 | Gancao | Glycyrin | 52.61 | 0.47 |
| MOL004882 | Gancao | Licocoumarone | 33.21 | 0.36 |
| MOL004883 | Gancao | Licoisoflavone | 41.61 | 0.42 |
| MOL004884 | Gancao | Licoisoflavone B | 38.93 | 0.55 |
| MOL004885 | Gancao | licoisoflavanone | 52.47 | 0.54 |
| MOL004891 | Gancao | shinpterocarpin | 80.3 | 0.73 |
| MOL004898 | Gancao | (E)-3-[3,4-dihydroxy-5-(3-methylbut-2-enyl)phenyl]-1-(2,4-dihydroxyphenyl)prop-2-en-1-one | 46.27 | 0.31 |
| MOL004903 | Gancao | liquiritin | 65.69 | 0.74 |
| MOL004904 | Gancao | licopyranocoumarin | 80.36 | 0.65 |
| MOL004907 | Gancao | Glyzaglabrin | 61.07 | 0.35 |
| MOL004908 | Gancao | Glabridin | 53.25 | 0.47 |
| MOL004910 | Gancao | Glabranin | 52.9 | 0.31 |
| MOL004911 | Gancao | Glabrene | 46.27 | 0.44 |
| MOL004912 | Gancao | Glabrone | 52.51 | 0.5 |
| MOL004913 | Gancao | 1,3-dihydroxy-9-methoxy-6-benzofurano[3,2-c]chromenone | 48.14 | 0.43 |
| MOL004914 | Gancao | 1,3-dihydroxy-8,9-dimethoxy-6-benzofurano[3,2-c]chromenone | 62.9 | 0.53 |
| MOL004915 | Gancao | Eurycarpin A | 43.28 | 0.37 |
| MOL004924 | Gancao | (-)-Medicocarpin | 40.99 | 0.95 |
| MOL004935 | Gancao | Sigmoidin-B | 34.88 | 0.41 |
| MOL004941 | Gancao | (2R)-7-hydroxy-2-(4-hydroxyphenyl)chroman-4-one | 71.12 | 0.18 |
| MOL004945 | Gancao | (2S)-7-hydroxy-2-(4-hydroxyphenyl)-8-(3-methylbut-2-enyl)chroman-4-one | 36.57 | 0.32 |

**Table 1** (*continued*)

| ID | Herb | Compound | OB(%) | DL |
|---|---|---|---|---|
| MOL004948 | Gancao | Isoglycyrol | 44.7 | 0.84 |
| MOL004949 | Gancao | Isolicoflavonol | 45.17 | 0.42 |
| MOL004957 | Gancao | HMO | 38.37 | 0.21 |
| MOL004959 | Gancao | 1-Methoxyphaseollidin | 69.98 | 0.64 |
| MOL004961 | Gancao | Quercetin der. | 46.45 | 0.33 |
| MOL004966 | Gancao | 3′-Hydroxy-4′-O-Methylglabridin | 43.71 | 0.57 |
| MOL004974 | Gancao | 3′-Methoxyglabridin | 46.16 | 0.57 |
| MOL004978 | Gancao | 2-[(3R)-8,8-dimethyl-3,4-dihydro-2H-pyrano[6,5-f]chromen-3-yl]-5-methoxyphenol | 36.21 | 0.52 |
| MOL004980 | Gancao | Inflacoumarin A | 39.71 | 0.33 |
| MOL004988 | Gancao | Kanzonol F | 32.47 | 0.89 |
| MOL004989 | Gancao | 6-prenylated eriodictyol | 39.22 | 0.41 |
| MOL004990 | Gancao | 7,2′,4′-trihydroxy—5-methoxy-3—arylcoumarin | 83.71 | 0.27 |
| MOL004991 | Gancao | 7-Acetoxy-2-methylisoflavone | 38.92 | 0.26 |
| MOL004993 | Gancao | 8-prenylated eriodictyol | 53.79 | 0.4 |
| MOL005000 | Gancao | Gancaonin G | 60.44 | 0.39 |
| MOL005001 | Gancao | Gancaonin H | 50.1 | 0.78 |
| MOL005003 | Gancao | Licoagrocarpin | 58.81 | 0.58 |
| MOL005007 | Gancao | Glyasperins M | 72.67 | 0.59 |
| MOL005008 | Gancao | Glycyrrhiza flavonol A | 41.28 | 0.6 |
| MOL005012 | Gancao | Licoagroisoflavone | 57.28 | 0.49 |
| MOL005016 | Gancao | Odoratin | 49.95 | 0.3 |
| MOL005017 | Gancao | Phaseol | 78.77 | 0.58 |
| MOL005018 | Gancao | Xambioona | 54.85 | 0.87 |
| MOL005020 | Gancao | dehydroglyasperins C | 53.82 | 0.37 |
| MOL000006 | Zhishi | luteolin | 36.16 | 0.25 |
| MOL001798 | Zhishi | neohesperidin_qt | 71.17 | 0.27 |
| MOL001803 | Zhishi | Sinensetin | 50.56 | 0.45 |
| MOL001941 | Zhishi | Ammidin | 34.55 | 0.22 |
| MOL002914 | Zhishi | Eriodyctiol (flavanone) | 41.35 | 0.24 |
| MOL005100 | Zhishi | 5,7-dihydroxy-2-(3-hydroxy-4-methoxyphenyl)chroman-4-one | 47.74 | 0.27 |
| MOL005828 | Zhishi | nobiletin | 61.67 | 0.52 |
| MOL005849 | Zhishi | didymin | 38.55 | 0.24 |
| MOL007879 | Zhishi | Tetramethoxyluteolin | 43.68 | 0.37 |
| MOL009053 | Zhishi | 4-[(2S,3R)-5-[(E)-3-hydroxyprop-1-enyl]-7-methoxy-3-methylol-2,3-dihydrobenzofuran-2-yl]-2-methoxy-phenol | 50.76 | 0.39 |
| MOL013277 | Zhishi | Isosinensetin | 51.15 | 0.44 |
| MOL013279 | Zhishi | 5,7,4′-Trimethylapigenin | 39.83 | 0.3 |
| MOL013430 | Zhishi | Prangenin | 43.6 | 0.29 |

**Table 1** (*continued*)

| ID | Herb | Compound | OB(%) | DL |
|---|---|---|---|---|
| MOL013435 | Zhishi | poncimarin | 63.62 | 0.35 |
| MOL013436 | Zhishi | isoponcimarin | 63.28 | 0.31 |
| MOL013437 | Zhishi | 6-Methoxy aurapten | 31.24 | 0.3 |
| MOL004828 | Gancao | Glepidotin A | 44.72 | 0.35 |
| MOL004829 | Gancao | Glepidotin B | 64.46 | 0.34 |
| MOL004833 | Gancao | Phaseolinisoflavan | 32.01 | 0.45 |
| MOL004835 | Gancao | Glypallichalcone | 61.6 | 0.19 |
| MOL004838 | Gancao | 8-(6-hydroxy-2-benzofuranyl)-2,2-dimethyl-5-chromenol | 58.44 | 0.38 |
| MOL004841 | Gancao | Licochalcone B | 76.76 | 0.19 |
| MOL004848 | Gancao | licochalcone G | 49.25 | 0.32 |
| MOL004849 | Gancao | 3-(2,4-dihydroxyphenyl)-8-(1,1-dimethylprop-2-enyl)-7-hydroxy-5-methoxy-coumarin | 59.62 | 0.43 |
| MOL004855 | Gancao | Licoricone | 63.58 | 0.47 |
| MOL004856 | Gancao | Gancaonin A | 51.08 | 0.4 |
| MOL000098 | Multi-herb | quercetin | 46.43 | 0.28 |
| MOL000354 | Multi-herb | isorhamnetin | 49.6 | 0.31 |
| MOL000359 | Multi-herb | sitosterol | 36.91 | 0.75 |
| MOL000422 | Multi-herb | kaempferol | 41.88 | 0.24 |
| MOL004328 | Multi-herb | naringenin | 59.29 | 0.21 |

quercetin ligand, the HCC target macromolecule, the hydrogen bond binding energy, total internal energy, and electrostatic energy are shown in Table 3. The inhibition constant is the dissociation constant of the protein-inhibitor complex and the inhibition constant is smaller as the inhibition strength increases. The molecular docking of CHEK2, BAX, AR, PTGS2 and quercetin are shown in Fig. 8.

## DISCUSSION

HCC arises from hepatocytes in the liver parenchyma and is typically seen in patients with chronic liver disease, including viral hepatitis, alcoholic liver disease, and nonalcoholic fatty liver disease (*Arslanoglu et al., 2016*). Hepatocarcinogenesis is a multi-factorial process with various signaling pathways that include the p53 signaling pathway, VEGF signaling pathway, TGF signaling pathway, and Ras MARK signaling pathway (*Grandhi et al., 2016*). 75% of patients with limited stage cancer have a survival rate of approximately 5 years when treated with curative therapies such as liver transplantation and surgical resection (*Guy et al., 2012*). However, for patients in hepatic decompensation or with a highly heterogeneous tumor, surgery is not always effective and has a significant economic burden on patients. HCC is resistant to chemotherapy, thus an alternative treatment must be established.

TCM is prepared according to the principle of "King, Vassal, Assistant, and Delivery servant" based on the patient's disease state (*Zhang et al., 2019*). This principle determines the major therapeutic compounds combined with other agents used to restore the balance of body functions and reduce toxicity, which can be exhibited in symptoms including

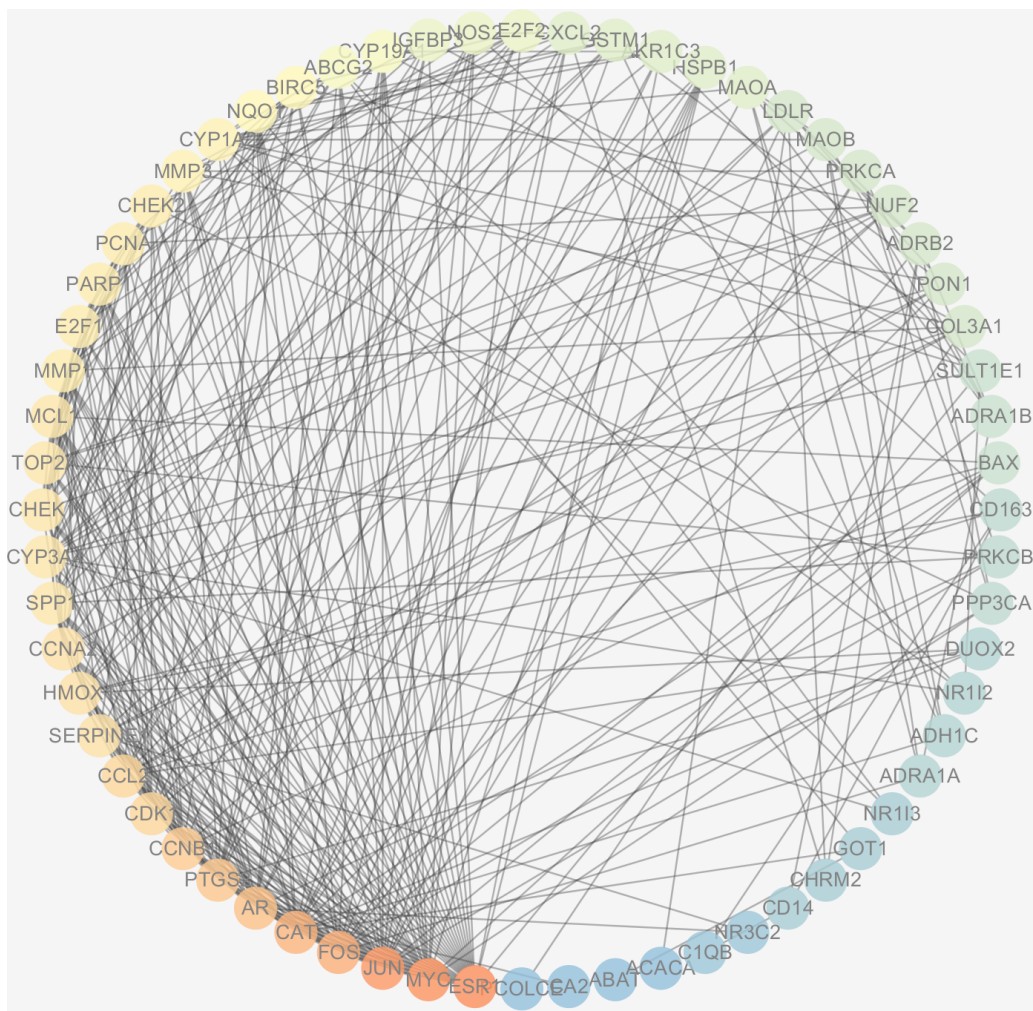

**Figure 6  The PPI network of the 64 targets of SiNiSan on stage III HCC.** The color of the nodes changing from blue to red represents the degree of the nodes changing from low to high.

insomnia, palpitation, infections, and fever (*Li et al., 2020a*; *Li et al., 2020b*). Studies have found that TCM plays an anti-tumor role through apoptotic pathways in ROS-mediated mechanisms, DNA damage-mediated mechanisms, Ca2+-mediated mechanisms, and by blocking transcription or translation (*Min, 2010*). TCM inhibits cell proliferation and promotes the apoptosis of tumor cells by targeting the stimulation of the host immune response for cytotoxic activity (*Liao et al., 2015*; *Zhao et al., 2018*). SNS has been used for centuries in China as a remedy for liver stagnation and spleen deficiencies. Recent experiments have shown that SNS is a remarkably effective treatment for hepatitis and liver injury (*Shu et al., 2018*). The individual herbs that make up SNS have anti-inflammatory, anti-bacterial, anti-oxidation, anti-tumor, anti-proliferation, anti-angiogenic, anti-HBV, and apoptotic effects in vitro and in vivo (*Zhao et al., 2018*; *Li et al., 2020a*; *Li et al., 2020b*; *Ma et al., 2014a*; *Ma et al., 2014b*; *Sun et al., 2019*; *Xiang et al., 2019*; *Pan et al., 2019*; *Zhou et al., 2019*; *Lefaki et al., 2020*).

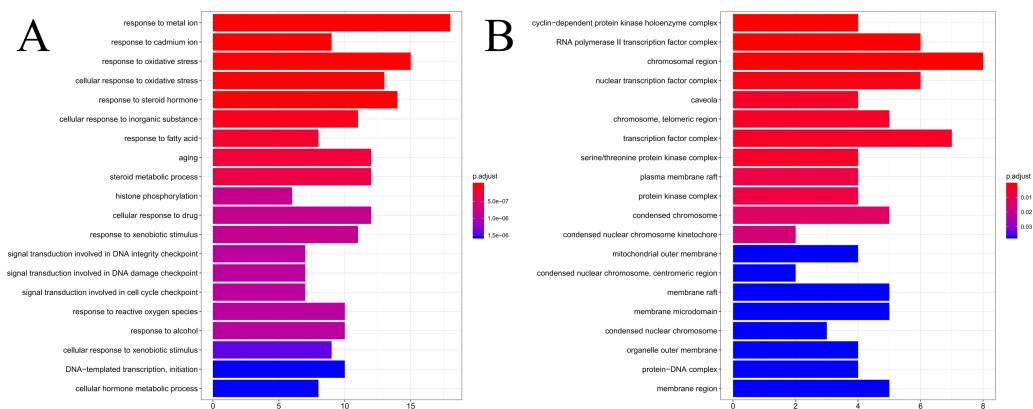

**Figure 7** **The results of GO enrichement analysis of the 64 targets of SiNiSan on stage III HCC.** (A) The top 20 biological processes (BPs); (B) the top 20 cell compounds (CCs).

**Table 2** **The results of KEGG pathway enrichment analysis of 64 intersection HCC genes.**

| ID | Description | pvalue | geneID |
|---|---|---|---|
| hsa04110 | Cell cycle | 4.70E−06 | CDK1/CCNA2/CHEK1/MYC/CCNB1/CHEK2/E2F1/E2F2/PCNA |
| hsa04115 | p53 signaling pathway | 9.62E−06 | BAX/CDK1/CHEK1/CCNB1/SERPINE1/CHEK2/IGFBP3 |
| hsa00982 | Drug metabolism - cytochrome P450 | 9.82E−05 | CYP3A4/CYP1A2/GSTM1/ADH1C/MAOB/MAOA |
| hsa00140 | Steroid hormone biosynthesis | 0.000424164 | CYP3A4/CYP1A2/AKR1C3/SULT1E1/CYP19A1 |
| hsa00350 | Tyrosine metabolism | 0.000521784 | ADH1C/MAOB/MAOA/GOT1 |
| hsa01522 | Endocrine resistance | 0.000536494 | BAX/JUN/ESR1/FOS/E2F1/E2F2 |
| hsa00360 | Phenylalanine metabolism | 0.000695885 | MAOB/MAOA/GOT1 |
| hsa00380 | Tryptophan metabolism | 0.000945976 | CYP1A2/CAT/MAOB/MAOA |
| hsa04020 | Calcium signaling pathway | 0.000947819 | ADRA1A/CHRM2/ADRA1B/ADRB2/PRKCA/NOS2/PPP3CA/PRKCB |
| hsa00330 | Arginine and proline metabolism | 0.00182834 | NOS2/MAOB/MAOA/GOT1 |
| hsa04064 | NF-kappa B signaling pathway | 0.004651344 | CD14/PTGS2/PRKCB/PARP1/CXCL2 |
| hsa04066 | HIF-1 signaling pathway | 0.005678296 | PRKCA/NOS2/HMOX1/PRKCB/SERPINE1 |
| hsa01524 | Platinum drug resistance | 0.007222453 | BAX/GSTM1/BIRC5/TOP2A |
| hsa00980 | Metabolism of xenobiotics by cytochrome P450 | 0.008704905 | CYP3A4/CYP1A2/GSTM1/ADH1C |

**Table 3  The results of molecular docking between quercetin and AR, BAX, CHEK2, ESR1, NOS2, PTGS2.** The results include binding energy, the inhibition constant, and the total internal energy(Vdw-hb-desolv + electrostatic energy) between the receptor and the ligand.

| Component ligand | Target | PDB ID | Hbonding atoms of target | Hbinding energy (Kcal/mol) | Inhib constant | Vdw-hb-desolv + electrostatic energy (Kcal/mol) |
|---|---|---|---|---|---|---|
| Quercetin | AR | 4WEV | GLU308:HN; ASP309:HN | −4.24 | 775.54 uM | −6.04 |
| Quercetin | PTGS2 | 5KIR | ASN43:H | −4.11 | 976.06 uM | −5.9 |
| Quercetin | PTGS2 | 5KIR | GLY66:H | −3.93 | 1.32 mM | −5.72 |
| Quercetin | AR | 2Q7K | ARG788:HH11; HIS789:HD1; GLN792:HE22 | −3.36 | 3.45 mM | −5.14 |
| Quercetin | CHEK2 | 2CN8 | ARG474:HH22 | −3.36 | 3.44 mM | −5.15 |
| Quercetin | PTGS2 | 5KIR | ARG77:HE; ARG77:HH11 | −3.2 | 6.28 mM | −4.79 |
| Quercetin | PTGS2 | 5KIR | ARG44:HE; CYS47:H | −3.19 | 4.55 mM | −4.99 |
| Quercetin | NOS2 | 5XN3 | GLU219:HN | −3.17 | 4.78 mM | −4.96 |
| Quercetin | PTGS2 | 5KIR | LYS83:HZ1 | −3.11 | 3.72 mM | −5.1 |
| Quercetin | AR | 2Q7K | ARG788:HH11; ARG788:HH21; HIS789:HD1; GLN792:HE22 | −3.05 | 5.8 mM | −4.85 |
| Quercetin | CHEK2 | 4A9U | ARG482:HE | −3.05 | 5.78 mM | −4.84 |
| Quercetin | CHEK2 | 2CN8 | LYS444:HZ1 | −3.04 | 5.86 mM | −4.84 |
| Quercetin | BAX | 4BD7 | SER60:HG; LYS64:HZ3 | −3.00 | 3.83 mM | −5.09 |
| Quercetin | PTGS2 | 5KIR | HIS90:HD1 | −2.91 | 6.7 mM | −4.75 |
| Quercetin | CHEK2 | 2CN8 | LYS464:HZ2 | −2.78 | 9.17 mM | −4.57 |
| Quercetin | AR | 5BXJ | ARG39:HH12; HIS56:HE2 | −2.23 | 23.17 mM | −4.02 |
| Quercetin | AR | 5BXJ | ARG39:HH22; LYS99:HZ1 | −2.19 | 24.99 mM | −3.98 |
| Quercetin | NOS2 | 4NOS | LYS123:HZ3 | −1.68 | 58.96 mM | −3.47 |
| Quercetin | BAX | 4BD6 | LYS119:HZ1 | −1.67 | 59.9 mM | −3.46 |
| Quercetin | ESR1 | 6KN5 | LYS996:HZ1 | −1.64 | 63.15 mM | −3.43 |
| Quercetin | BAX | 4BD6 | SER72: HG | −1.59 | 68.29 mM | −3.38 |
| Quercetin | ESR1 | 4XI3 | LYS467:HZ1 | −1.59 | 68.65 mM | −3.53 |

In our research, there are 64 overlaps between drug targets and disease targets which are the important targets involved in the therapeutic mechanism of SNS, we studied 113 compounds and 64 target genes using the screening criteria and found that each bioactive compound acts on at least 1 HCC differential gene, core targets genes were PTGS2, ESR1, AR , NOS2, CCNA2 and CHEK1. The majority of the compounds were derived from Radix Glycyrrhizae (Gancao), which indicated that Radix Glycyrrhizae may be the principal active herb in treating HCC. Radix Bupleuri, Fructus Aurantii Immaturus, and Radix Paeoniae Alba may play a supporting role. Quercetin and kaempferol are bioactive ingredients with the highest number of targets and both coexist in multi-herb formulations (quercetin from Radix Glycyrrhizae and Radix Bupleuri, kaempferol from Radix Glycyrrhizae, Radix Bupleuri and Radix Paeoniae Alba). In order to study the molecular mechanism of disease development from a systematic perspective, we constructed a PPI network to predict the mode of action between HCC-related proteins and to provide a reference for the role of drugs. AR and ESR1 were found to be the targets that interact with the most proteins in the PPI network and also act as targets for a large number of drug components. This suggests that AR and ESR1 may be important nodes for SNS to act on HCC. AR is an

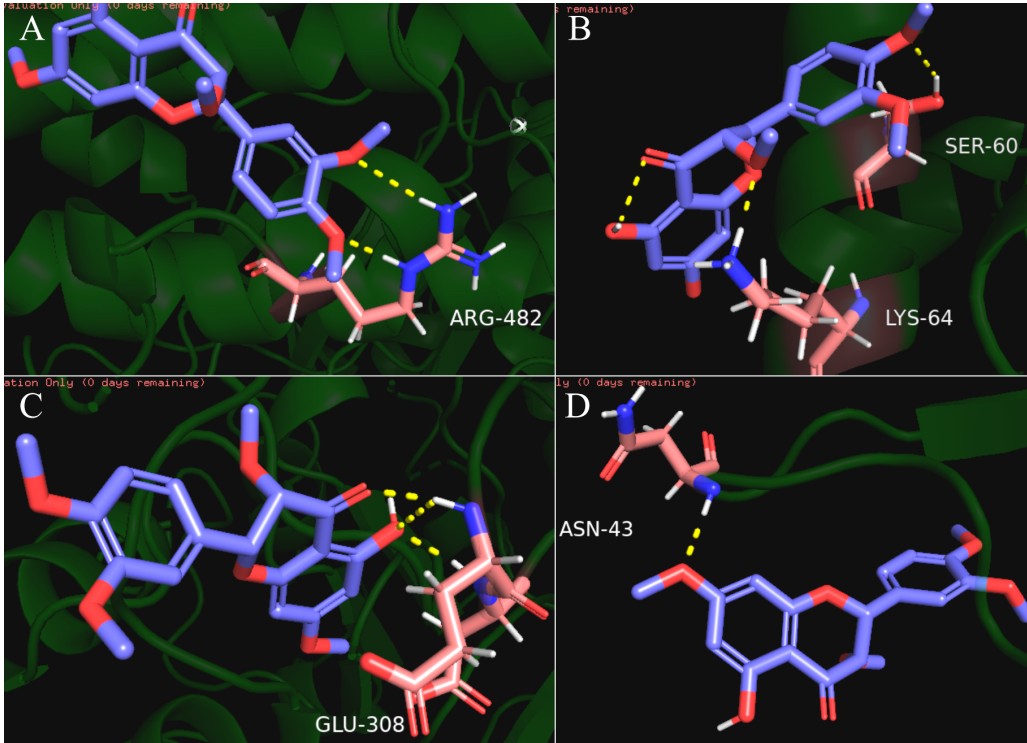

**Figure 8** **The results of molecular docking between quercetin and genes CHEK2, BAX, AR, PTGS2.**
The blue molecule is quercetin, the pink is the amino acid residue where the target protein is docked with quercetin, and the yellow dotted line is the hydrogen bond. (A) the molecular docking between quercetin ligand and CHEK2 macromolecule, the amino acid residue forming a hydrogen bond with quercetin is ARG-482; (B) the molecular docking between quercetin ligand and BAX macromolecule, the amino acid residue forming a hydrogen bond with quercetin is LYS-64 and SER-60; (C) the molecular docking between quercetin ligand and AR macromolecule, the amino acid residue forming a hydrogen bond with quercetin is GLU-308; (D) the molecular docking between quercetin ligand and PTGS2 macromolecule, the amino acid residue forming a hydrogen bond with quercetin is ASN-43.

androgen receptor involved in the growth and progression of hepatocellular carcinoma, it can inhibit the expression of CD90 in circulating tumor cells by up-regulating histone 3H2A (*Han et al., 2020*). Targeting the AR transduction pathway may be the key to inhibiting the progression of advanced HCC (*Han et al., 2020*; *Ma et al., 2014a*; *Ma et al., 2014b*). ESR1 is an estrogen receptor that causes abnormal serum methylation and can be used as a biomarker for the diagnosis of HCC. The imbalance of ESR1 expression can cause a high degree of HCC amplification (*Tsiambas et al., 2011*; *Dou et al., 2016*) and is a focus of targeted therapy.

Enrichment analysis revealed that these compounds may act against HCC through: the cellular response to steroid hormones; oxidative stress; histone phosphorylation; steroid metabolic processes; cyclin-dependent protein kinase holoenzyme complex; synthesis of RNA polymerase II transcription factor complex and the nuclear transcription factor complex. The response of these biological functions on steroid hormones coincided with the target AR and ESR1, suggesting that SNS may regulate the levels of sex hormones.

Histone H3 phosphorylation is involved in the chemical carcinogenesis of HCC, while histone 4 phosphorylation is related to the proliferation and differentiation of hepatocytes after a liver injury (*Sukumar et al., 2020*; *Liu & Liu, 2020*). The potential pathways are: the p53 signaling pathway; calcium signaling pathway; NF-kappa B signaling pathway; and the HIF-1 signaling pathway. The P53 pathway has the highest degree of enrichment, combined with the previous gene set GSEA analysis. The P53 pathway is also significantly enriched in the overall stage III hepatocellular carcinoma gene set. Previous studies have demonstrated that P53 inhibits the progression, recurrence, and related immune responses of HCC (*Besant & Attwood, 2013*; *Xiaonian et al., 2017*). The signaling pathway exists in p53 in HCC based on the HCC map in KEGG (https://www.kegg.jp) and acts as tumor suppressor. The SNS target genes in the p53 pathway include BAX, CDK1, CHEK1, CCNB1, SERPINE1, CHEK2, and IGFBP3. Quercetin interferes with 6 of the 7 target genes in the above-mentioned p53 pathway (BAX, CDK1, CCNB1, SERPINE1, CHEK2, IGFBP3) and molecular docking confirmed that quercetin can bind to the P53 pathway-related targets (CHEK2, SPRPINE1, BAX). Thus, quercetin is likely to be a promising HCC treatment drug in SNS. Quercetin is a bioflavonoid with high antitumor activity that acts on cancers such as osteosarcoma and colon cancer by rebuilding the tumor microenvironment and alleviates liver damage by regulating the EGFR pathway (*Choi et al., 2010*; *Lan et al., 2017*; *Hu et al., 2017*; *Carrasco-Torres et al., 2017*; *Massi et al., 2017*), Quercetin can exert anti-tumor activity by targeting HSP70/90, P13K, and NF-B and has been reported in the treatment of hepatocellular carcinoma (*Fernández, Fondevila & Méndez, 2019*; *Wang et al., 2011*). Our other derived pathways require testing to determine their efficacy in the treatment of HCC.

## CONCLUSIONS

TCM assumes the characteristics of multiple compounds, targets and pathways when applied to disease prevention and treatment, which is the focus of network pharmacology. We predicted the targets and potential mechanisms of SNS compounds for HCC treatment by constructing networks and used molecular docking to simulate their behavior. We focused on quercetin's targeting of genes that were enriched in the p53 pathway and then inhibited the progression of HCC by regulating the p53 pathway. GSEA enrichment analysis of the HCC gene set and the KEGG enrichment analysis of the intersection gene after the differential analysis show that the P53 pathway is a pathway with significant influence. AR and ESR1 in HCC targets are also worthy of attention due to their large number of SNS components and the proteins that are highly interactive with other proteins in the PPI network. Research has revealed that these two sex hormone receptors are involved in the development of HCC. SNS can inhibit HCC by affecting hormone levels and by regulating the P53 pathway. The results of this study should be verified by in vivo and vitro experiments and more studies are needed to reduce the toxicity of TCM and prove its value in the treatment of HCC.

## ACKNOWLEDGEMENTS

We would like to thank Professor Helin Feng for the research guidance and technical support of this study.

### Funding

The authors received no funding for this work.

### Competing Interests

The authors declare there are no competing interests.

### Author Contributions

- Qin Zhang conceived and designed the experiments, performed the experiments, analyzed the data, prepared figures and/or tables, authored or reviewed drafts of the paper, and approved the final draft.
- Zhangying Feng conceived and designed the experiments, performed the experiments, prepared figures and/or tables, authored or reviewed drafts of the paper, and approved the final draft.
- Mengxi Gao conceived and designed the experiments, analyzed the data, prepared figures and/or tables, and approved the final draft.
- Liru Guo conceived and designed the experiments, analyzed the data, authored or reviewed drafts of the paper, and approved the final draft.

### Data Availability

Data is available at NCBI GEO: GSE101685.

### Supplemental Information

Supplemental information for this article can be found online at http://dx.doi.org/10.7717/peerj.10745#supplemental-information.

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
