# Peer review of "Determining novel candidate anti-hepatocellular carcinoma drugs using interaction networks and molecular docking between drug targets and natural compounds of SiNiSan"

_PeerJ, doi:10.7717/peerj.10745_

## Round 0.1 · original submission · Major Revisions

Your manuscript has now been reviewed and the reviewers' comments are appended below. I agree with them in finding overall your manuscript interesting however, they have raised several concerns that need to be addressed by a major revision. All the reviewers' comments are pertinent and should be addressed in detail in the revision. Moreover, I ask the authors to get editing help from a professional English language correction service or colleagues with full professional proficiency in English.

·

Basic reporting

Zhang et al. investigated the main functional molecules in SiNiSan and their potential targets by a network-based polypharmacology strategy. The authors recommended some targets in the p53 pathway, however, there is no in vitro and in vivo data to support their hypotheses.

Experimental design

no comment

Validity of the findings

no comment

Additional comments

Some concerns should be addressed before its publication in this journal.

1. Figure 3: The figure legend needs to be revised because there is no “Black cube” in the network.
2. Figure 4: What do the yellow nodes represent?
3. Figure 7:
The figures are very ugly, and should better be re-rendered.
The figure legend needs to be revised because the P53 pathway-related targets listed are not consistent with ones studied by molecular docking.
In the main text, the binding modes of quercetin to the selected targets should better be discussed with more details. Intermolecular interactions apart from the hydrogen bonding interactions also need to be discussed. Besides, if there are common binding patterns of quercetin to the three selected targets?
The docking energies to the three selected targets are too small, which suggests that quercetin is unlikely to bind. The authors should better add the unit for the docking energy.
4. Quercetin is a well-known multi-target drug, and the reported targets of quercetin need to be discussed.
5. Tables 1 and 2: Why the table titles are duplicate?
6. There are lots of spelling and grammar errors. For example, the presentation of the name of prof. Feng in Acknowledgements is wrong.

·

Basic reporting

(1) The language in this manuscript is ambiguous, and the conclusion is exaggerated. Please see Major Q1 and Q7.
(2) There are some problems in the figures. Please see Minor Q1 to Q3.
(3) Raw data are supplied.

Experimental design

(1) The experimental method has many problems and needs to be redesigned. Please see Major Q2 to Q7.
(2) There is no problem with the statistical method in this manuscript.

Validity of the findings

(1) The findings could not be verified. Please see Major Q1.

Additional comments

Hepatocellular carcinoma (HCC), the most common form of primary liver cancer, typically develops within the background of chronic liver disease. It is an aggressive disease with a dismal prognosis. Therefore, it is necessary to develop more effective therapies. Sinisan, a traditional Chinese medicine, is effective for the treatment of gastrointestinal disorders. In this study, Dr. Qin Zhang and colleagues investigated the potential mechanism of SNS for the therapy of HCC by constructing interaction networks and molecular docking. The study provides the conclusion, which showed that quercetin in SNS may treat HCC by targeting on SERPINE1, CCNB1 and CDK1 gene in p53 signaling pathway. There are some problems in this study, which require the author to revise the research design, and there are some major concerns that need to be addressed.

Major concerns:
1. “Validity of the findings & Introduction” Line 95-99. “It has been believed to be effective for treating liver stagnation and spleen deficiency, improving disorders of digestive system and alleviating mental depression (Jiang et al., 2003). While the unclear mechanism of TCM has hampered its clinical practice in the world, the therapeutically effect of SNS on Hepatocellular Carcinoma remains to be further elucidated.” The purpose of this study was to analyze and predict the potential active components and potential targets and mechanisms of SiNiSan against HCC by using the pharmacobioinformatics method. The significance of this study is based on the premise that SiNiSan has significant anti-liver cancer activity. However, according to the literature investigation described by the author and myself, there is no clear evidence that SiNiSan is clinically used in anti-liver cancer therapy, nor has it been found to have significant anti-liver cancer activity in animals. Therefore, the existing experimental data in this manuscript cannot support the conclusion of the study. The author must verify the anti-liver cancer activity of SiNiSan by adding in vivo and in vitro experiments. The first method can directly verify the anti-liver cancer activity of SiNiSan by using the transplanted tumor model of mice, or the second method can confirm the anti-liver cancer activity of the active components of SiNiSan screened in this study by using in vitro methods.
2. “Materials & Methods – Screening of Bioactive compounds and targets in SiNiSan” Line 124. “Set the filter criteria as OB≥30% and DL≥0.18 for potential bioactive compounds…” What is the rationale for selecting these two screening criteria? What are the references? The authors must elaborate in the text.
3. “Results – Bioactive compounds and targets in SiNiSan” Line 169-174. “In all, 113 compounds were identified as potential bioactive molecules for further study.” What is the content of these 113 compounds in SiNiSan? The effect of the drug depends on its activity and content, and it is hard to believe that all 113 compounds are effective. The authors should indicate the content of these 113 compounds in SiNiSan using experimental or database data.
4. “Result – Hepatocellular Carcinoma-associated targets” Line 176-178. Why choose the GSE101685 data set with few samples? The expression profile of HCC with different pathological types, clinical stages and carcinogenic factors is different. Therefore, the author should select a specific type or clinical stage of HCC. It is recommended that the authors use the TCGA data set, taking into account patient risk factors (HBV, HCV or alcohol) and clinical stages.
5. “Result – Analysis of Protein- protein interactions (PPI) network” Line 191-195 and Figure 4. In this part of the experimental results, the author established the interaction network of differentially expressed genes of HCC. However, the author did not elaborate on the relationship between this part of the experimental content and the theme of this study. This makes the description of the results in this section less logical.
6. “Result – GO and KEGG pathway enrichment analysis and KEGG network” Line 197-205 and Figure 5 and 6. In these results, the effect of SNS on HCC by GO and KEGG pathway enrichment analysis and KEGG network were analyzed, but the authors did not specify which genes were used for analysis.
7. “Result – Docking results of compound and core target” Line 207-216 and Figure 7. In this part, the author studied the potential direct effects of quercetin and SPRPINE1, CDK2, CCNB2, and IGFBP3 proteins using docking. But there are a lot of problems here. First of all, quercetin is an active ingredient widely distributed in Traditional Chinese Medicine and has been studied the most. Therefore, quercetin has the most identified targets, and using Quercetin as a key component of SiNiSan clearly lacks logic. Second, molecular docking is used to study the potential direct interactions between molecules, and only those molecular interactions that interact with the pharmacodynamics pocket may be biologically active. The authors do not explain the impression of molecular docking conformation on the biological conformation of proteins, so these data are meaningless. Third, AutoDock is a semi-flexible docking software, which requires the crystal structure of interaction between positive drugs and target proteins to be obtained first. Based on the drug action pocket, the screening docking is conducted by AutoDock. Obviously, the authors did not obtain the corresponding crystal structure, and did not use a reasonable docking method, but simply docking. Generally speaking, this part of data lacks verifiability and practical significance.

Minor comments:
1. Figure 4. What does the yellow node represent?
2. Figure 5. Please ensure the clarity of the text.
3. Figure 6. Please add the x-coordinate title.

Reviewer 3 ·

Basic reporting

NA

Experimental design

NA

Validity of the findings

NA

Additional comments

In this manuscript "A strategy to find novel candidate anti-hepatocellular carcinoma drugs by constructing interaction networks and molecular docking between drug targets and natural compounds of SiNiSan formula" Zhang et al. have analyzed a traditional Chinese medicine -SiNiSan which has been used as a treatment strategy for liver and spleen related diseases. They analyzed the active compounds in SiNiSan and performed various bioinformatics analysis and commented on quercetin and its target genes. Although the question of understanding the application of Chinese medicine to hepatocellular cancer patients sounds interesting however manuscript in it’s current form is very poorly written and lacks novel insights gained from the bioinformatics analysis.

Specific comments that need to be addressed are listed below:

1. TCGA has a huge dataset of hepatocellular cancer patients. It is not clear why authors have confined their analysis to very few patient sample data from GEO to identify differentially expressed genes in normal vs. cancer patients. What pathways these differentially expressed genes represents? A GSEA can help in understanding the core genes/pathways contributing to hepatocellular cancer in patients. Figure2: Does distinct clusters shows different pathways? It is also not clear how the expression data was normalized.

2. Figure3: what is the biological significance of this network analysis? What does different colors represent? Figure legends should clearly explain the figure. Why this analysis was restricted to only top differentially expressed genes? A detailed functional interpretation of such complex network can aid in understanding the biological significance of these differentially expressed and target genes.

3. What was the criterion used to call a gene as differentially expressed? Did authors perform any normalization of gene expression data downloaded from GEO? It should be clearly mentioned in the methods section of the manuscript.

4. It is not clear what authors meant by preliminary screening network (line 194)? What does DC>61 and BC>600 represents? What does yellow node represents in Figure 4? What are the biological insights gained from this protein-protein interaction data analysis?

5. Please provide all the technical details (including parameters) of how autodock was performed for the docking analysis. What is the rationale for performing a docking experiment here? Does quercentin has significantly higher binding affinity to SPRPINE1, CDK2, CCNB2 as compared to any other protein? Line 212-217 what is protein A, protein E and protein C?

6. What is the overlap of bioactive compounds and targets among various individual herbs in SNS? Do these targets belong to same distinct pathways?

7. Figure 1: It is not clear what authors meant by OB+DL screening. Abbreviations should be limited in the figures for better understanding. "Differential anglysis" -> Differential analysis Abbreviation used in figure should be clearly described in figure legend.

8. Figure quality is very poor and needs significant improvement.

9. For the screening of bioactive compounds in Chinese herbal medicine database how filtering criteria as OB≥30% and DL≥0.18 were identified for potential bioactive compounds identification.

10. Figure 5: what does colors indicates?

11. Figure6: what does the x-axis represents?

---

## Round 0.2 · Major Revisions

Your manuscript has now been reviewed and the reviewers' comments are appended below. The manuscript improved but there are still several major concerns that need to be addressed with additional data analysis and clearer explanation.

·

Basic reporting

(1) The language in this manuscript has improved.
(2) Picture quality has improved.
(3) Raw data are supplied.

Experimental design

(1) The experimental method has some problems and needs to be redesigned. Please see Major Q2 to Q3.
(2) There is no problem with the statistical method in this manuscript.

Validity of the findings

(1) The findings of this manuscript should be verifiable.

Additional comments

Hepatocellular carcinoma (HCC), the most common form of primary liver cancer, typically develops within the background of chronic liver disease. It is an aggressive disease with a dismal prognosis. Therefore, it is necessary to develop more effective therapies. Sinisan, a traditional Chinese medicine, is effective for the treatment of gastrointestinal disorders. In this study, Dr. Qin Zhang and colleagues investigated the potential mechanism of SNS for the therapy of HCC by constructing interaction networks and molecular docking. The study provides the conclusion, which showed that the SNS component has a large number of stage III HCC targets. Among the targets, the sex hormone receptors, the AR and ESR1 genes, are the core targets of SNS component and the most active proteins in the PPI network. In addition, quercetin, which has the most targets, can act on the main targets (BAX, CDK1, CCNB1, SERPINE1, CHEK2, and IGFBP3) of the P53 pathway to treat HCC. Although the author answered all of the questions, I still have some questions about the responses, which need further answers from the authors.

Major concerns:
1. “Introduction” The authors need to clearly explain why the differentially expressed genes for stage III HCC were analyzed.
2. “Prediction and Gene Set Enrichment Analysis (GSEA) enrichment analysis of stage III hepatocellular carcinoma targets” Line 134-147. In the GSEA analysis, how did the authors group the samples and then identify the target genes associated with stage III HCC? The author needs to explain in detail.
3. “Differential analysis of stage III hepatocellular carcinoma targets and the building of a compound-disease target network” Line 150-157. Same as major Q2, which two or more groups were compared to obtain differentially expressed genes?
4. “Response to Major Q1” The supplement of in vivo and in vitro experiments is my suggestion to the author, which is the most direct way to increase the significance of this study. I can understand that the authors do not want to add too much workload, but I do not suggest the authors to use the epidemic situation as a reason. As we all know, the COVID19 epidemic situation in mainland China has been well controlled in May.

Reviewer 3 ·

Basic reporting

NA

Experimental design

NA

Validity of the findings

NA

Additional comments

Fig1: A bar plot representing the enriched termed at a fdr threshold will be helpful as compared to Figure 1A. What does yellow region represent?

How much is the overlap between GSEA enriched terms in GEO and TCGA datasets?

Figure3: What are the top genes? Please mark them for clear understanding

Figure4: A statistical analysis of significance of intersection of targets of SNS component and HCC genes will be helpful as compared to non informative network representation. Similarly what is the significance of PPI analysis as shown in Figure 5

---

## Round 0.3 · accepted · Accept

Your manuscript has now been reviewed again by one of our reviewers. In light of this advice, we are happy to publish this suitably revised version of your manuscript in PeerJ.

·

Basic reporting

No comment.

Experimental design

No comment.

Validity of the findings

No comment.

Additional comments

Hepatocellular carcinoma (HCC), the most common form of primary liver cancer, typically develops within the background of chronic liver disease. It is an aggressive disease with a dismal prognosis. Therefore, it is necessary to develop more effective therapies. Sinisan, a traditional Chinese medicine, is effective for the treatment of gastrointestinal disorders. In this study, Dr. Qin Zhang and colleagues investigated the potential mechanism of SNS for the therapy of HCC by constructing interaction networks and molecular docking. The study provides the conclusion, which showed that the SNS component has a large number of stage III HCC targets. Among the targets, the sex hormone receptors, the AR and ESR1 genes, are the core targets of SNS component and the most active proteins in the PPI network. In addition, quercetin, which has the most targets, can act on the main targets (BAX, CDK1, CCNB1, SERPINE1, CHEK2, and IGFBP3) of the P53 pathway to treat HCC. The paper is improved and most concerned raised by the reviewer have been addressed. I think it is might suitable for publication at this version of revised manuscript.